# mRNA Therapeutic Vaccine for Hepatitis B Demonstrates Immunogenicity and Efficacy in the AAV-HBV Mouse Model

**DOI:** 10.3390/vaccines12030237

**Published:** 2024-02-25

**Authors:** Dorien De Pooter, Wim Pierson, Soheil Pourshahian, Koen Dockx, Ben De Clerck, Isabel Najera, Heather Davis, Ellen Van Gulck, Daniel Boden

**Affiliations:** 1Infectious Diseases Discovery, Janssen Research & Development, LLC, a Johnson & Johnson Company, Turnhoutseweg 30, 2340 Beerse, Belgium; wpierso1@its.jnj.com (W.P.); bdeclerc@its.jnj.com (B.D.C.); hdavis12@its.jnj.com (H.D.); evangulc@its.jnj.com (E.V.G.); 2RNA and Targeted Therapeutics, Johnson & Johnson Innovative Medicine, 1600 Sierra Point Parkway, Brisbane, CA 94005, USA; 3Charles River Laboratories, Turnhoutseweg 30, 2340 Beerse, Belgium; kdockx@its.jnj.com; 4Infectious Diseases Discovery, Janssen Research & Development, LLC, a Johnson & Johnson Company, 1600 Sierra Point Parkway, Brisbane, CA 94005, USA; inajera@its.jnj.com (I.N.); dboden@mail.com (D.B.)

**Keywords:** chronic hepatitis B, therapeutic vaccination, mRNA vaccine, lipid nanoparticles, AAV-HBV mice, HBsAg reduction

## Abstract

Chronic infection with hepatitis B virus (HBV) develops in millions of patients per year, despite the availability of effective prophylactic vaccines. Patients who resolve acute HBV infection develop HBV-specific polyfunctional T cells accompanied by neutralizing antibodies, while in patients with chronic hepatitis B (CHB), immune cells are dysfunctional and impaired. We describe a lipid nanoparticle (LNP)-formulated mRNA vaccine, optimized for the expression of HBV core, polymerase, and surface (preS2-S) antigens with the aim of inducing an effective immune response in patients with CHB. Prime and prime/boost vaccination with LNP-formulated mRNA encoding for core, pol, and/or preS2-S dosing strategies were compared in naive C57BL/6 and BALB/c mice. Immune responses were assessed by IFN-γ ELISpot, intracellular cytokine staining (ICS), and ELISA for antibody production, whereas anti-viral efficacy was evaluated in the AAV-HBV mouse model. The mRNA vaccine induced strong antigen-specific polyfunctional T cell responses in these mouse models, accompanied by the emergence of anti-HBs and anti-HBe antibodies. After three immunizations, the antigen-specific immune stimulation resulted in up to 1.7 log_10_ IU/mL reduction in systemic HBV surface antigen (HBsAg), accompanied by a transient drop in systemic HBeAg, and this was observed in 50% of the AAV-HBV-transduced mice in the absence of additional modalities such as adjuvants, HBsAg reducing agents, or checkpoint inhibitors. However, no treatment-related effect on viremia was observed in the liver. These results warrant further optimization and evaluation of this mRNA vaccine as a candidate in a multimodal therapeutic regimen for the treatment of chronic HBV infection.

## 1. Introduction

Despite the long standing availability of effective prophylactic vaccines, around 300 million people worldwide are still affected by chronic hepatitis B infection [1]. After an acute phase, most adults but only few infants resolve their infection; and those who cannot become chronically infected, typically for life, with a greatly elevated risk of developing liver cirrhosis and hepatocellular carcinomas [2,3,4]. The current standard of care for chronic hepatitis B infection (CHB) is the long-term use of nucleoside analogs (Nas) for the suppression of viral replication [5]. However, these NAs do not target the covalently closed circular DNA (cccDNA) that is formed in the nucleus of infected hepatocytes [6]. The persistence of this cccDNA leads to the constant uncontrolled production of viral antigens and viral replication, causing immune tolerance and contributing to the chronic HBV liver pathogenesis [7].

Therefore, life-long therapy is necessary to suppress viral resurgence and to decrease the progression of liver disease [8]. An alternative therapy is pegylated interferon alpha (PEG-IFN-α), which is typically administered for 48-weeks and can induce a loss of hepatitis B surface antigen (HBsAg) in circulation in up to 10% of the treated patients [9,10]. This therapy is known to be accompanied by unpleasant side effects [11]. There is thus an unmet need for a therapy that leads to the functional cure of chronic hepatitis B defined as a sustained, undetectable level of systemic HBsAg and HBV DNA after treatment is discontinued [12]. 

Patients with CHB are most often infected at birth (vertical transmission) or early in life. Their immune cells in the liver become exhausted due to the persistent presence of HBV antigens [13]. In this state of exhaustion, the T cells are weak: they lack virus-specific reactivity, defined as reduced cytotoxicity, decreased cytokine production, and a persistent expression of inhibitory receptors such as programmed cell death-1 (PD-1) and cytotoxic T lymphocyte-associated antigen-4 (CTLA-4) [14]. Most people infected with HBV in adulthood do resolve the acute HBV infection, and this is associated with the induction of HBV-specific polyfunctional CD4^+^ and CD8^+^ T cells and neutralizing antibodies that can kill infected hepatocytes [15]. 

A single therapy that can break tolerance, reinvigorate T cells, and bring about functional cure has not yet been discovered; however, the induction of HBV-specific CD4^+^ and CD8^+^ T cell responses with therapeutic vaccination is believed to be a promising avenue towards achieving functional cure. Previously, our group developed a plasmid DNA (pDNA) vaccine encoding for HBV antigen core and polymerase (pol), which induced robust immune responses in naive mice and non-human primates. Despite the induction of a strong T cell response in naive mice, the vaccine did not affect viral parameters after a prime/boost dosing regimen in the AAV-HBV mouse model [16]. This pDNA vaccine was also evaluated in healthy volunteers and CHB-infected patients where HBV-specific immune responses were generated that were lower and more restricted in CHB patients compared to healthy volunteers [17]. This may result from immune tolerance, which is believed to be caused by the persistent exposure to high levels of HBV antigens, envelope in particular [18,19]. Including surface antigen (envelope) in the vaccine might thus be essential to create HBsAg-specific T cells and antibodies that can target the infected hepatocytes as the number of envelope-specific T cells in CHB patients is low [18,20]. 

Although DNA vaccines are stable, easy to store, and have been shown to induce immune responses in humans, electroporating high doses is required, and even then, responses are not as strong as with other platforms. Breaking immune tolerance using a DNA vaccine is considered to only be achieved in combination with an adjuvant [21]. The development of highly effective mRNA vaccines against SARS-CoV-2 has shown that this vaccine platform is well tolerated and can induce strong immune responses, including T cells [22,23]. These immune responses are elevated compared to DNA vaccines since they only need to enter the cytoplasm while DNA needs to reach the nucleus [24]. Moreover, mRNA is encapsulated in a lipid nanoparticle (LNP) which can serve as both a delivery system and an adjuvant [25].

This manuscript describes the immune responses and anti-viral effects in mice of a therapeutic HBV vaccine composed of LNP-formulated mRNA, designed based on the previously described pDNA vaccine but including the surface antigen to also create HBsAg-specific T cells and antibodies to target infected hepatocytes to increase the low number of such T cells in CHB patients [16,20,26]. The vaccine consists of three different mRNA strands, each encoding for one HBV antigen: core, pol, or envelope (specifically preS2-S) (Figure 1a). The mRNAs were first formulated individually and evaluated for their immunogenicity in naive mice (Figure 1b) [27,28]. Given the advantage for further development, the mRNAs were also evaluated admixed (Figure 1c) and co-encapsulated (Figure 1d) in both naive and AAV-HBV-transduced mice. The LNPs used so far are MC3-or ALC-0315-based.

## 2. Materials and Methods

### 2.1. Generation and LNP Formulation of mRNA

The mRNA was synthesized using the HiScribe^®^ T7 in vitro transcription kit (NEB, Ipswich, MA, USA) from linearized DNA template where ribonucleotide uridine was replaced with N1-Methylpseudoridine-m1ΨTP (TriLink Biotechnologies, San Diego, CA, US). It contained 5′ and 3′ untranslated region (UTR), polyA and was capped using CleanCap^®^ Reagent AG (TriLink Biotechnologies, San Diego, CA, USA). RNA was purified using miniprep/maxiprep kits (Qiagen, Hilden, Germany) according to the manufacturer’s protocol. 

The selected genes of interest (GOIs) encode for HBV core (pre-core not included), pol, and surface (preS2.S) antigens preceded by open reading frames. The antigen sequences are consensus sequences from HBV genotypes A, B, C, and D while retaining the most common HLA-A, HLA-B, and HLA-C epitopes. The multi-genotype HBV sequences were codon-optimized for optimal expression in humans. 

The lipid nanoparticle used for formulation with the mRNA consisted of an ionizable lipid 4-(dimethylamino)-butanoic acid, (10Z,13Z)-1-(9Z-12Z)-9,12octadecadien-1-yl-10,13-nonacedacien-1-yl-ester (MC3; custom synthesized), or 4-(hydroxybutyl-azanediyl)di(hexane-6,1-diyl) bis(2-hexyldecanoate) (ALC-0315, Avanti Polar Lipids, Alabaster, AL, USA). In addition, cholesterol, 1,2-distearoyl-sn-glycero-3-phophocholine (DSPC), and 1,2-dimyristoyl-rac-glycero-3-methoxypolyethylene glycol-2000 (DMG-PEG2000) (all from Avanti Polar Lipids) were used in the formulation. A solution containing ALC-0315 or MC3, DSPC, and DMG-PEG2000 at a molar ratio of 46.3:9.4:42.7:1.6 (ALC-0315) or 50:10:38.5:1.5 (MC3) was prepared in ethanol (Millipore Sigma, Burlington, MA, USA). The mRNA was suspended in sodium acetate (50 mM, pH 4.5), which was then combined with the lipid mixture in a microfluidic mixer (NanoAssamblr^®^ Ignite™, Precision NanoSystems, Vancouver, BC, Canada) at a flow rate of 3:1 and N/P molar ratio of 6:1. Amicon filters (MWCO 100 kDa; Millipore Sigma, Burlington, MA, USA) were used for sample buffer exchange to 20 mM Tris-HCl pH 7.5. Sucrose (Millipore Sigma, Burlington, MA, USA) was added to the samples (final concentration 10% *w*/*v*) before filtration through a 0.2 µm filter. All formulations were stored at −80 °C. Formulations were tested for particle size (60–80 nm) and encapsulation efficiency (>90%). All formulations were stored at −80 °C.

### 2.2. In Vivo

All mice used in these experiments were female C57BL/6 or BALB/c mice obtained from Janvier (Mayenne, France).

#### 2.2.1. Immunization of Naïve Mice

C57BL/6 or BALB/c mice (6–8 weeks old) were immunized with LNP-formulated mRNA (the dose was experiment-dependent; see relevant figures) via intramuscular (IM) injection in the gastrocnemius (50 µL). In a prime only experiment, vaccination was performed on day 0 and splenocytes were harvested on day 14. For prime/boost experiments, vaccination was performed on days 0 and 21 with the endpoint at day 28. Mice in the control group were dosed with saline.

#### 2.2.2. Immunization of AAV-HBV-Transduced Mice

Female C57BL/6 mice (6–8 weeks old) were transduced with 2.5 × 10^9^ viral genome equivalents (vge) of rAAV-HBV1.3-mer WT replicon (genotype D) (BrainVTA, Wuhan, China) per mouse via the tail vein. Immunization was initiated 77 days after HBV transduction by intramuscular (IM) injection in the gastrocnemius with 50 µL of vaccine containing 10 µg of core, pol, and preS2.S (3.3 µg of each mRNA) at weeks 0, 3, and 6 (n = 8) [29]. The negative control group was dosed with an equal volume of saline.

Blood leucocytes were collected 2 weeks before the start of treatment, 1 week post each dose, and at the end of the experiment, to perform IFN-γ ELISpot. Serum was collected weekly throughout the study and stored at −80 °C until used to measure viral parameters and/or antibodies. 

At the end of the study, part of the liver was used for the isolation of resident intrahepatic lymphocytes (IHL) to assess the expression of exhaustion markers using multicolor flow cytometry staining. The rest of the liver was formalin-fixed and embedded in paraffin (FFPE) to visualize HBsAg and HB core antigen (HBcAg) expression and infiltrating CD3 T cells using immunohistochemistry (IHC). The spleen was also harvested to perform multicolor flow cytometry staining to determine the polyfunctionality of HBV-specific T cells. 

### 2.3. Ex Vivo

#### 2.3.1. Viral Parameters, Antibodies, and Alanine Aminotransferase (ALT) Analyses

Serum HBsAg, HBeAg, anti-HBe, and anti-HBs concentrations were measured using CLIA kits (Ig Biotechnology, Burlingame, CA, USA) according to the manufacturer’s protocol. Dilutions in PBS were adapted depending on the estimated levels. Plate read-out was performed with Viewlux ultra HTS microplate imager (Perkin Elmer, Mechelen, Belgium) [30]. ALT levels were analyzed using 4 µL of serum, with a commercial kit according to the manufacturer’s guidance (Sigma-Aldrich, Saint Louis, MO, USA) and a Spark multimode microplate reader (Tecan, Mechelen, Belgium) [30].

#### 2.3.2. Isolation of Splenocytes

Spleens were isolated and tissue dissociation was performed using a GentleMACS Octo Dissociator (Miltenyi Biotec, Bergisch Gladbach, Germany); red blood cells were lysed using ACK lysis buffer (Lonza, Basel, Switzerland) for 5 min. The cells were washed twice after which viability and cell concentrations were determined using a Cellaca MX Cell Counter (Nexcelom, MA, USA). Splenocytes were tested at 100,000 cells/well on IFN-γ ELISpot plates (Mabtech, Nacka Strand, Sweden), and 1 million cells were used per flow cytometry panel.

#### 2.3.3. Isolation of Blood Leucocytes

Whole blood was transferred from the capillary tubes to 5 mL tubes over 70 µm cell strainers; the samples of two mice were pooled and depleted of red blood cells by using 200 µL anti-Ter119 MicroBeads. Magnetic isolation was carried out using a QuadroMACS separator following the manufacturer’s protocol (Miltenyi Biotec) [30]. Cell viability and concentration were counted using a Cellaca MX Cell Counter (Nexcelom), followed by performing IFN-γ ELISpot using 100,000 leucocytes.

#### 2.3.4. Detection of HBV-Specific T Cells by IFN-γ ELISpot

Cells were stimulated with overlapping peptide pools covering the sequence for core, pol, or preS2.S (JPT, Gladbach, Germany) during overnight incubation on pre-coated IFN-γ ELISpot plates to identify the number of IFN-γ secreting cells. Due to the large size of pol, its peptide pool was split in two pools, pol1 and pol2, that were tested separately. Dimethyl sulfoxide (DMSO) was used as negative control, to subtract background signal. ELISpot was performed following the manufacturer’s guidelines (Mabtech, Nacka Strand, Sweden). Results were expressed as the number of spots forming cells per million, mean ± standard deviation.

#### 2.3.5. Measurement of HBV-Specific Polyfunctional T Cells 

Cells were stimulated with overlapping peptide pools covering the sequence for core, pol (2 pools), and preS2.S (JPT, Gladack, Germany) during a 6 h incubation period to identify the number of IFN-γ- and TNF-α-secreting cells. To identify these cells, intracellular staining was performed by first staining the dead cells (fixable viability dye eFluor780, 65-0865-18, eBioscience, San Diego, CA, USA). After washing away the dye, the cells were blocked using Fc (anti-CD16/CD32, 553142, BD Biosciences, Frankin Lakes, NJ, USA) and True Stain Monocyte blocker (426103, Biolegend, San Diego, CA, USA). After washing, the cells were fixated and permeabilized using the Fix/Perm solution kit (554714, BD Biosciences). The cells were washed and an antibody cocktail in Perm buffer (BD Biosciences) was added to identify CD4^+^ (BUV395, 563790, BD Biosciences), CD8^+^ (BV786, 563332, BD Biosciences), and CD3^+^ T cells (PerCp-Cy5.5, 560527, BD Biosciences) that secrete IFN-γ (PECF594, 562303, BD Biosciences) and TNF-α (APC, 506308, Biolegend). Data were acquired with a LSRFortessa flow cytometer (BD Biosciences), and analysis was performed using FlowJo version v10.8.1 (BD Biosciences). DMSO was used as negative control, and background signal was subtracted from the resulting proportion of IFN-γ-TNF-α double positive cells. Leucocyte Activation Cocktail (LAC, 550583, BD Biosciences) was used as experimental positive control. The gating strategy used to determine the antigen-specific IFN-γ and TNF-α double positive CD4^+^ and CD8^+^ T cells can be found in Appendix A.

#### 2.3.6. Immunohistochemistry (IHC) 

IHC staining for HBcAg and HBsAg and image analysis were performed as described in a previous paper [30].

#### 2.3.7. Statistical Analysis

Statistical comparisons were performed using GraphPad Prism 9. All *p*-values ≤ 0.05 were considered significant (* *p* ≤ 0.05, ** *p* ≤ 0.01, *** *p* ≤ 0.001, and **** for *p* ≤ 0.0001). Details of the tests are provided with each figure. 

## 3. Results

The formulated mRNA therapeutic vaccine with three strands encoding for each of the HBV antigens was evaluated as (i) individually formulated mRNAs, (ii) a coformulation of the three individual mRNA-LNPs, and (iii) a co-encapsulation of three mRNA strands in one LNP (Figure 1). 

### 3.1. Low Doses of LNP-Formulated mRNA Encoding for Single HBV Antigen Induced Strong Immune Responses in Naive C57BL/6 Mice

The IFN-γ ELISpot responses in naive mice immunized once with LNP-formulated mRNA encoding core (Figure 2a) were dose-dependent, with 1 µg being the lowest dose able to induce a significant IFN-γ response compared to saline control (104 ± 59.5 versus 3.70 ± 4.08 IFN-γ spot forming cells (SFC)/million cells, *p* < 0.0001) (Figure 2b). Responses increased proportionally with higher doses of 2 µg and 5 µg without reaching a plateau at the highest tested dose (352 ± 239 IFN-γ SFC/million cells). 

When a prime/boost regimen was tested at a 5 µg dose (Figure 2a), core-specific recall responses were elevated, but not significantly increased compared to post-prime levels (prime 277 ± 249 versus prime/boost 858 ± 595 IFN-γ SFC/million of cells, *p* = ns) (Figure 2c). Subsequent prime/boost vaccination using three dose levels of mRNA (2, 5, and 10 µg) did not significantly increase responses to core either, as they already plateaued with the lowest dose of 2 µg (Figure 2d; 10 µg: 1349 ± 640; 5 µg: 1013± 465; and 2 µg: 1543 ± 825 IFN-γ SFC/million cells). 

The prime/boost strategy after the dosing of LNP-formulated mRNA encoding for pol also elevated responses compared to prime only (Figure 2c). This increase was significant for pol1 (2233 ± 828 versus 921 ± 406 IFN-γ SFC/million cells, *p* = 0.0018). For pol2, levels reached the upper limit of quantification (ULOQ) after boosting for 50% of the mice, which was not the case for prime only. In contrast to core, the 10 µg dose did result in a higher number of SFCs compared to the 5 µg dose for both pol1 (2278 ± 1163 versus 949 ± 583 IFN-γ SFC/million cells; *p* = 0.031) and pol2, where four out of six mice showed levels that reached ULOQ for the highest dose (Figure 2d). Lowering the dose to 2 µg did not result in changes compared to the 5 µg dose for pol1 (1401 ± 514 IFN-γ SFC/million cells) or pol2 (3197 ± 1240 SFC/million cells). It needs to be stated that for both pol1 and pol2, there was some study-variability as the responses to a 5 µg dose in Figure 2d are lower compared to the responses measured in the experiment shown in Figure 2c (pol1: 949 ± 583 versus 2233 ± 828; pol2: 2354 ± 591 versus 4486 ± 671 IFN-γ SFC/million cells). 

The preS2-S-specific recall responses reached ULOQ after prime/boost dosing, while this was not the case for prime only of formulated mRNA encoding for preS2-S at 5 µg (Figure 2c). However, this response level was already observed at the lowest dose of 2 µg after prime/boost and also at 10 µg (Figure 2d).

These results show that the range of vaccine-induced responses was antigen-dependent, with core responses being detected with as little as a single dose of 1 µg. Prime/boost vaccination increased the magnitude of responses to all antigens, with maximum responses already induced with 2 µg of mRNA, except for pol. 

### 3.2. Coformulated mRNAs Encoding for Core, Pol, and preS2-S Induce Polyfunctional Responses and Antibody Production

A vaccine for human use would ideally have all the antigens coformulated to allow for a single injection, and thus this approach was tested in mice to see how responses compare to when the antigens were tested individually. mRNAs encoding for each of the antigens were individually formulated into LNPs and then combined at doses of 1, 2, or 5 µg of each antigen expressing mRNA. Prime/boost regimen of total mRNA doses of 3, 6, or 15 µg were tested in naive C57BL/6 mice, eliciting responses to all antigens. The preS2-S recall responses exceeded maximum detectable levels at all dose levels, and no dose effect was observed for core-specific responses. A dose–response was observed, however, with pol where the 15 µg dose gave stronger pol1 responses than the 3 µg dose (448 ± 179 versus 153 ± 75.5, *p* = 0.035) (Figure 3a). Responses to pol2 were significantly greater with 15 µg than with 3 µg (687 ± 193 versus 2328 ± 734, *p* < 0.0001) as well as with 6 µg (988 ± 714 versus 2328 ± 734, *p* = 0.0002). Of note, it was observed that for the coformulation, the magnitude of pol-specific induced responses decreased significantly compared to the responses for the single-antigen vaccine (Appendix A). Core-specific responses were not significantly impacted, and for preS2-S, this could not be assessed as values reached ULOQ.

A dose-dependent induction of anti-HBsAg antibodies (anti-HBs) was observed on day 28 in all mice dosed with 15 µg, but not with the lower doses (Figure 3b). Anti-HBeAg antibodies (anti-HBe) were induced in all mice for all doses, one week post boost (Appendix A).

T cell polyfunctionality, as defined by CD4^+^ or CD8^+^ expressing T cells which secrete both IFN-γ and TNF-α, was investigated, with findings in line with the results shown by the ELISpot read-out: the proportion of core-specific polyfunctional CD8^+^ T cells was low (<0.06%), and no dose-related effects could be demonstrated. Core-specific CD4+ T cell polyfunctionality was not impacted by doses either (15 µg: 0.32 ± 0.15; 6 µg: 0.32 ± 0.14; and 3 µg: 0.39 ± 0.08%, respectively). This was different for pol2 where the 15 µg dose induced more CD8^+^ polyfunctional T cells compared to 6 µg (*p* = 0.0212) and 3 µg (*p* = 0.0088) (Figure 3c). However, there was no dose-dependent effect on the proportion of CD4^+^ polyfunctional T cells (Figure 3d). Interestingly, for preS2-S, an inverse dose–response was observed, where the lowest dose induced the highest polyfunctional CD8^+^ T cell responses (Figure 3c), whereas no dose-dependent effect could be detected on the proportion of CD4^+^ polyfunctional T cells (Figure 3d). Taken together, these data show that the multi-antigen vaccine was able to induce HBV-specific antibodies and polyfunctional T cells; responses were slightly lower compared to when dosing with the single-antigen-encoding mRNA for core, pol1, and pol2.

### 3.3. Minor Differences in Response between Co-Encapsulation and Coformulation of Three mRNAs

In the studies discussed in the previous section, the three mRNAs had been LNP-formulated separately before they were mixed (coformulation) (Figure 1c). In this study, coformulated multi-antigen vaccine (5 µg mRNA per antigen) was compared to a co-encapsulated mRNA vaccine in which the three mRNAs were mixed prior to LNP formulation (Figure 1d). 

In naive C57BL/6 mice, the co-encapsulated mRNAs induced better T cell responses for core (1397 ± 482 versus 721 ± 319 IFN-γ SFC/million cells, *p* = 0.0177), for pol1 (1594 ± 454 versus 978 ± 731 IFN-γ SFC/million cells, *p* = ns), and for pol2 than the coformulated mRNAs (Figure 4a). For preS2-S, both approaches gave responses at ULOQ, so no conclusion could be drawn. 

Both coformulation and co-encapsulation induced a similar level of antibodies (Figure 4b,c): comparable levels of anti-HBs were observed on day 28 (443 ± 746 mIU/mL with four out of eight responders versus 818 ± 1462 mIU/mL with five out of eight responders) and for anti-HBe on day 14 post prime (coformulation 2.27 ± 0.83 versus co-encapsulation 2.35 ± 1.075 PEIU/mL; five out of eight responders for both groups).

Similarly, the equivalent induction of polyfunctional pol and preS2-S CD8^+^ and CD4^+^ T cells was observed after the administration of co-encapsulated and coformulated mRNA vaccine (Figure 4d,e). The proportion of core-specific polyfunctional CD8+ T cells was low for both coformulated and co-encapsulated mRNA (0.026 ± 0.043 and 0.029 ± 0.019% IFN-γ TNF-α double positive cells). Polyfunctional CD4^+^ T cells were induced for all HBV antigens without differences between the two strategies. 

All together, these data showed only minor differences between the co-encapsulation and coformulation approaches of the multi-antigen mRNA vaccines. 

### 3.4. The Strong Vaccine-Induced Response Was Confirmed in the BALB/c Mouse Strain

The afore mentioned results were all generated in a C57BL/6 mouse strain. The co-encapsulated vaccine was also evaluated using the prime/boost strategy (Figure 2a) in BALB/c mice that differ from C57BL/6 in their major histocompatibility complex (MHC) since they contain haplotype H2b instead of H2d [31]. 

The magnitude of the induced response after therapeutic vaccination following the prime/boost schedule, as depicted in Figure 2a, was evaluated by IFN-g ELISpot. At the end of the study (day 28), responses were observed for all antigens: core- and pol2-specific recall responses were higher compared to saline control (core: 527 ± 159 versus 1.30 ± 1.79 IFN-γ SFC/million splenocytes; *p* = 0.0006 and pol2: 430 ± 408 versus 5.74 ± 5.84 IFN-γ SFC/million splenocytes; *p* = ns), whereas pol1 and preS2-S induced specific recall responses that were all above the upper limit of quantification of the assay (Figure 5a).

In two out of eight mice, detectable concentrations of antibodies against envelope could be measured in serum two weeks post prime (Figure 5b). Three weeks post prime, at the time of the second dose, this was the case in 50% of the animals. At the end point (day 28), concentrations of anti-HBs antibodies in serum were elevated in all animals, with concentrations exceeding maximum levels of detection for half of the mice. HBe antibodies were also measured at the end of the experiment, with concentrations in sera exceeding maximum detection limit for all mice (Appendix A). There was no induction of HBe antibodies at the earlier time points.

As described in C57BL/6 mice, HBV-specific polyfunctional T cells were also induced in BALB/c mice after vaccination, although with a different pattern for pol1- and pol2-specific responses across the mouse strains. The proportion of polyfunctional CD8^+^ T cells (Figure 5c) was comparable after 6 h of pol1 and preS2-S peptide stimulation, with a significant difference compared to saline control (pol1 4.86 ± 1.92 versus 0.013 ± 0.012; preS2-S 5.81 γ 1.64 versus 0.013 ± 0.0085, with *p* < 0.0001 for both). There was also an HBV-specific polyfunctional CD4^+^ T cellular response (Figure 5d) for all HBV antigens (*p* < 0.0001 for core, pol1, and preS2-S; and *p* = 0.0042 for pol2), albeit lower compared to CD8^+^ T cell responses and to those observed in C57BL/6 mice.

### 3.5. Co-Encapsulated mRNA Vaccine Achieves up to 1.7 Log_10_ Reduction in HBsAg Levels in the AAV-HBV-Transduced Mouse Model

In order to evaluate whether the observed HBV-specific polyfunctional T cells and antibodies in naive mice could also be induced in the context of HBV antigenemia, and whether vaccination could reduce viral load and HBsAg levels, AAV-HBV mice (on C57BL/6 background) with baseline HBsAg levels about 10^3^ IU/mL were immunized with a 10 µg total dose of the formulated multi-antigen vaccine (1:1:1 weight ratio of antigens mRNA) on day 0, 21, and 42 (Figure 6a). In previous studies with the pDNA therapeutic vaccine, declines in HBsAg levels were shown when animals were vaccinated four times [30]. Given the higher level of immunogenicity from the mRNA vaccine in naive mice compared to pDNA [16], it was decided to administer the mRNA vaccines three times instead of four.

HBV-specific immune responses of blood leucocytes measured one week post each dose were induced for all antigens. Core showed the lowest IFN-γ SFC/million cells (Figure 6b), which was also observed in naive C57BL/6 mice (Figure 3a). After every immunization, an increase in responses was observed for all antigens, which was followed by a steep decline after treatment. The induced T cell responses were accompanied by antibodies against HBsAg and HBeAg, starting after the last dose (Figure 6c). 

These responses induced a reduction in HBsAg concentrations between 1 and 1.7 log_10_ IU/mL in four out of eight mice at the end of the study compared to pre-vaccination levels (Figure 6d). A decline in HBsAg levels was observed after the last immunization in seven out of eight mice, with one mouse showing a 1 log_10_ IU/mL reduction in HBsAg already after the first dose. 

Regarding HBeAg, only immediately after the last immunization was there a transient drop between 0.4 and 0.7 log_10_ IU/mL in four out of eight mice. These mice correspond to the ones that had a decline in HBsAg levels. However, at the end of the study, none of the mice showed a significant reduction in HBeAg concentration compared to baseline levels (Figure 6e). Although effects were limited to a maximum drop of 1.7 log_10_ in HBsAg concentrations and did not reach LLOQ for the four responders and the HBeAg drop was only transient, these data show that the vaccine was able to induce HBV-specific polyfunctional T cells and antiviral antibodies in an AAV-HBV-transduced mouse model with a reducing effect on HBsAg levels.

To understand what changes the vaccine might be inducing, ALT levels were determined and immunohistochemistry (IHC) staining on liver tissue was performed (Figure 7a), confirming the presence of viral antigens in the liver (Figure 7c). ALT levels (Figure 7b), measured over time during the experiment (Figure 6a), did not show any treatment-related effect between the vaccinated mice and the control group, indicating no or little clearance of infected hepatocytes. 

Effects on systemic HBsAg levels were seen for 50% of the mice, and a limited drop in the proportion of HBsAg-positive cells in the liver (Figure 7c) was detected after vaccination; however, no complete loss was observed, and no significant treatment-related effect compared to the saline control group (5.49 ± 3.98 versus 3.16 ± 1.91% positive surface area, *p* = ns). No difference in the proportion of HBcAg-positive cells between saline and vaccinated mice (16.06 ± 6.21 versus 14.89 ± 4.10% positive cells, *p* = ns) was observed at the given total dose of 10 µg. This correlates with the drop in HBeAg levels in the serum being only transient (Figure 6d). (IHC images for the individual mouse can be found in Appendix A for HBsAg staining for saline- and mRNA-treated mice, respectively, and Appendix A for HBcAg for saline- and mRNA-treated mice, respectively.)

The generation of vaccine-induced HBV-specific polyfunctional T cells was also observed in the spleen of naive mice (Figure 3c,d and Figure 4c,d) as well as in the AAV-HBV-transduced mice (Figure 7d,e). The results showed that HBV-specific polyfunctional CD8^+^ T cells were generated with highest responses for pol (*p* = 0.0131 and <0.0001 for pol1 and pol2 compared to the saline control group). The same observation was true for HBV-specific polyfunctional CD4^+^ T cells but with lower proportions compared to CD8^+^ T cells (*p* = 0.0166 for pol2 compared to saline control group). Overall, the proportion of induced polyfunctional T cells, CD8^+^ and mainly CD4^+^, was reduced compared to the levels induced in naive mice. The mice with a reduction in HBsAg levels were mainly high in preS2-S-specific polyfunctional CD8^+^ T cells.

## 4. Discussion

Despite decades of research and scientific advancements, it is still a challenge to develop regimens providing functional cure in a substantial proportion of CHB patients. It is now generally accepted that therapeutic regimens consisting of compounds with different mechanisms of action will be required for achieving functional cure. Therapeutic vaccination for inducing HBV-specific T cells is considered to be an essential contributor to such a multimodal regimen, if combined with potent antiviral approaches that can reduce or remove the immune tolerizing effects of HBV antigens, such as HBsAg. 

Our previously described pDNA vaccine showed better immunogenicity in naive mice and healthy volunteers than in the presence of HBV antigenemia, namely, in the AAV-HBV mouse model and in patients with CHB, respectively [16,17]. In contrast to this pDNA vaccine [16], the mRNA vaccine described in this manuscript encodes not for two but for three HBV antigens, adding the envelope antigen (preS2-S) to core and pol. The inclusion of the HBV envelope in a vaccine design is believed to be important since envelope-specific T cells are generally low or not detectable in CHB patients, especially later in the disease course [18,20,26]. A recently published study, where our pDNA vaccine was supplemented with pDNA encoding for envelope (S), did show a small decline in HBsAg in AAV-HBV-transduced mice [16,30]. Now, an LNP-formulated mRNA platform was tested using this multi-antigen approach, including envelope. This platform should show enhanced immunity due to the superior in vivo transfection compared to the indispensable electroporation of pDNA, and the intrinsic immunostimulatory properties of both the mRNA and LNP [32]. 

It is believed that core-specific T cell responses are instrumental for antiviral outcome in CHB. It was shown that the core antigen-specific T cell response was associated with the resolution of chronic HBV infection in a patient who received a bone marrow transplant from a donor that cleared acute HBV infection [33]. It was also described that HBV resolvers show elevated levels of core- and pol-specific responses, which are associated with long-term virological control compared to those individuals that relapse after standard therapy discontinuation [34]. We saw that the multi-antigen vaccine significantly reduced core-specific CD4^+^ and CD8^+^ responses in AAV-HBV mice when compared to naive mice. Altering the antigen ratio of the vaccine might increase the core-specific responses in the AAV-HBV-transduced model.

Other than the selection and ratio of the mRNA encoding for HBV antigens, increased immune responses and the efficacy of the vaccine after IM injection might be obtained using an LNP formulation designed for the optimal delivery of mRNA into myocytes or antigen-presenting cells, such as dendritic cells, within the draining lymph nodes [35,36]. MC3 and ALC-0315 lipid-based formulations have been used so far, where MC3-LNP was originally designed for siRNA delivery to the liver and ALC-0315-based LNP was developed for the COVID-19 prophylactic vaccines [37]. Even though the generated immune responses are robust, the optimization of the LNP could increase the delivery of the mRNA to dendritic cells in the muscle or draining lymph nodes, which should lead to an improved antigen presentation. This could be achieved by adjusting the surface charge of the nanoparticles [38,39]. Improvements in delivery to immune cells have been made by increasing the size of LNPs by the addition of salt or by using piperazine-containing ionizable lipids [39,40,41].

The magnitude of mRNA vaccine-induced responses was lower in AAV-HBV-transduced mice compared to naive mice, which aligns with our previously published findings using the pDNA vaccine in mice and humans [16,17]. However, the results show that vaccination can achieve a reduction in systemic HBsAg of >1 log_10_, as observed in the AAV-HBV mouse model, with a limited effect in the liver as ALT levels were not increased, indicating there is a lack of or only a limited number of infected hepatocytes killed. The HBsAg levels measured in the liver were slightly decreased for the mice with lowered systemic HBsAg levels, but there was no complete loss in the liver. Both observations indicate that liver tolerance was not overcome. The drop in systemic HBsAg levels suggests that with adequate induced immunity in CHB patients, a control of viremia might be possible, especially if the vaccine is combined with an antiviral approach to first lower HBsAg. Clinical data to date suggest that lowering HBsAg levels is likely necessary but not sufficient to overcome HBV-specific T cell exhaustion and achieve functional cure. The high-level expression of checkpoint receptors in T cells is similarly observed in the context of oncology patients and therapeutic vaccine research [42]. Nonetheless, there is growing recognition that novel approaches targeting the viral lifecycle and supporting long-term immune control hold promise for potential curative therapies. 

Combination therapies of different modalities are therefore thought to be essential in achieving functional cure for CHB [43,44]. The ideal combination needs to be determined experimentally; however, it is believed that therapeutic vaccination should be complemented with treatments to reduce immune tolerance by first lowering HBsAg levels, such as HBV-targeted siRNA, antisense oligonucleotide (ASO), or HBsAg monoclonal antibodies [18,23,45,46,47]. In the AAV-HBV-transduced mouse model, we and others have shown promising results using siRNA and therapeutic vaccination, determined by sustained HBsAg loss [30,48]. Furthermore, this approach has already been evaluated in the clinic, where a vaccine therapy did maintain low HBsAg levels below 100 IU/mL for 33 out of 34 patients, accompanied by enhanced HBV-specific T cell responses after the HBsAg level was initially reduced with siRNA for a duration of 24 weeks [49]. Other immunotherapies to support therapeutic vaccination could be envisaged, such as Toll-like receptor agonists to improve host innate immune response against hepatitis B or by the use of checkpoint inhibitors [18,44,50,51].

Tempering the HBV-specific immune tolerance, combined with vaccination, should ideally induce both humoral and cellular responses in which both CD4^+^ and CD8^+^ T cells are thought to be critical. The drop in CD4^+^ T cell responses that we observed in AAV-HBV mice compared to naive mice could potentially be addressed with a heterologous prime/boost vaccination strategy, where the prime induces a stronger CD4^+^ T cell-directed primary response, using, for example, protein/peptide vaccines and/or CD4^+^ directed adjuvants, followed by a stronger CD8^+^ T cell-mediated booster secondary response, using, for example, an mRNA vaccine [52,53,54,55]. Using prime vaccination with HBV S and core protein formulated with potent adjuvants, followed by a modified vaccinia Ankara (MVA) boost, Protzer et al. showed that priming CD4^+^ T cell response is essential to reach a meaningful outcome in the therapeutic vaccination of AAV-HBV mice, as the CD4^+^ response helps in the decline of HBsAg levels and vaccine-induced CD8^+^ T cells clear infected cells [48,56]. The importance of CD4^+^ T cell engagement was confirmed by another study that demonstrated the requirement of CD4^+^ T cells to maintain HBsAg suppression using T cell depletion experiments in AAV-HBV mice [56]. The success of a heterologous prime/boost scheme, with a CD4^+^ directed prime and the mRNA vaccine described here as booster, needs to be determined experimentally. 

## 5. Conclusions

A therapeutic vaccine in the form of LNP-formulated mRNA encoding for HBV antigens was tested for its ability to induce an HBV-specific immune response in different mouse models. Key findings are that strong immune responses, as are required to overcome CHB, were induced in naive C57BL/6 and BALB/c mice, while somewhat reduced responses were observed in AAV-HBV-transduced C57BL/6 mice, a surrogate mouse model for chronic HBV. However, after three immunizations with the mRNA vaccine, a reduction in HBsAg serum levels between 1 and 1.7 log_10_ was achieved in half of the immunized mice population; and this was in the absence of any additional modalities such as HBsAg-reducing agents, adjuvants, or checkpoint inhibitors. It will be interesting to see how the mRNA vaccine will perform as a component of a multimodal therapeutic strategy for HBV cure.

## Figures and Tables

**Figure 1 vaccines-12-00237-f001:**
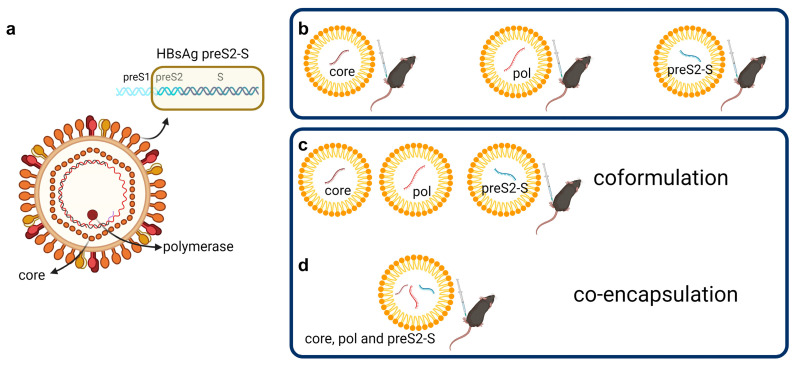
**Vaccine composition** (**a**) The vaccine is composed of mRNA encoding for antigens of the HBV virion: core, polymerase, and preS2-S envelope protein; (**b**) LNPs containing mRNA encoding for a single HBV antigen; (**c**) the coformulated mRNA vaccine is a mixture of the three different LNPs, each containing mRNA encoding for a single HBV antigen; (**d**) the co-encapsulated formulation contains mRNA encoding for all antigens, encapsulated within one LNP. (One mRNA copy per LNP is for illustration only; created with BioRender.com, accessed on 19 January 2024.)

**Figure 2 vaccines-12-00237-f002:**
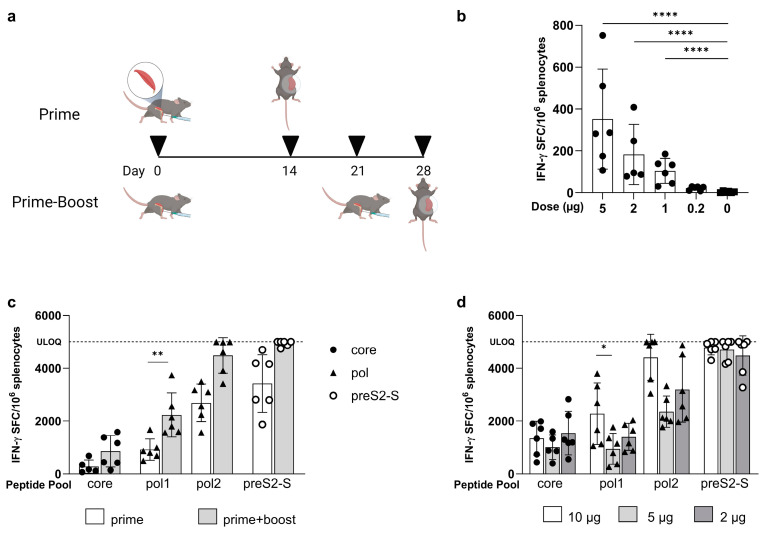
**Evaluation of induced immune responses in naïve C57BL/6 mice after vaccination with MC3-formulated mRNA vaccine encoding for one HBV antigen**. (**a**) Dosing scheme for prime and prime/boost strategy. Mice that were only primed received a vaccine on day 0 where responses are evaluated on day 14. In a prime/boost strategy, mice received a prime (day 0) and a boost (day 21) with an evaluation of the responses on day 28 (created with BioRender.com, accessed on 19 January 2024). (**b**) Induced immune responses, as measured by IFN-γ ELISpot (vertical axis), in a dose range (horizontal axis) finding (5, 2, 1, and 0.2 µg) after a prime dose of core-expressing mRNA (n = 6). (**c**) Induced immune responses, as measured by IFN-γ ELISpot (vertical axis), split per peptide stimulus (horizontal axis) in a comparison between prime (white) and prime/boost (grey) dosing schedules for the dosed formulated mRNA core (circles), pol (triangles), and preS2-S (open circles) at 5 µg (n = 6). (**d**) Induced immune responses, as measured by IFN-γ ELISpot (vertical axis), split per peptide stimulus (horizontal axis) in a comparison between different doses (10 µg white, 5 µg light grey, and 2 µg dark grey) of formulated mRNA core (circles), pol (triangles), and preS2-S (open circles) in a prime/boost dosing strategy (n = 6). Results are shown as mean ± standard deviation. Statistical analysis is performed by one-way ANOVA of the mean of log_10_ transformed data points (* *p* ≤ 0.05, ** *p* ≤ 0.01, ****, *p* ≤ 0.0001).

**Figure 3 vaccines-12-00237-f003:**
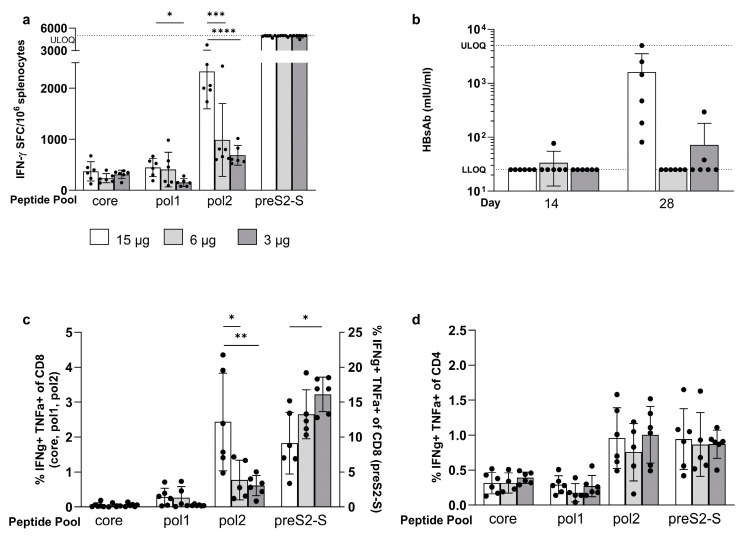
**Evaluation of dose finding study in naive C57BL/6 mice after vaccination with MC3-coformulated mRNAs**. Naive C57BL/6 mice (n = 6) were dosed via prime/boost strategy using 15 µg (white bars), 6 µg (light grey bars), and 3 µg (dark grey bars) of coformulation encoding for the HBV antigens. (**a**) Induced immune responses, as measured by IFN-γ ELISpot (vertical axis), split per peptide stimulus (horizontal axis). (**b**) Induced concentrations of anti-HBs (vertical axis), as measured by CLIA, on day 14 and 28 (horizontal axis). % of polyfunctional CD8^+^ T cells (**c**) and CD4^+^ T cells (**d**) defined as % of IFN-γ and TNF-α double positive cells of parent (vertical axis), split per peptide stimulation (horizontal axis). Results are shown as mean ± standard deviation. Statistical comparison is performed by unpaired *t*-test or one-way ANOVA on log_10_ transformed data for ELISpot and on logit transformed data for ICS (* *p* ≤ 0.05, ** *p* ≤ 0.01, *** *p* ≤ 0.001, and **** for *p* ≤ 0.0001).

**Figure 4 vaccines-12-00237-f004:**
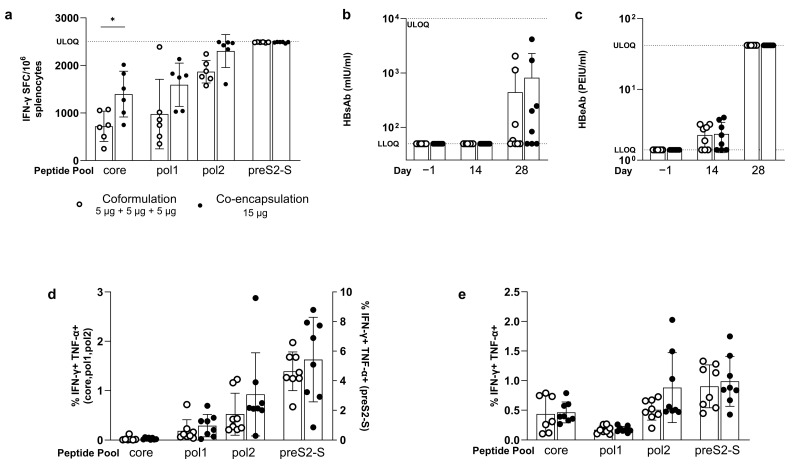
Evaluation of induced immune responses in naive C57BL/6 mice after vaccination with coformulated versus co-encapsulated (MC3) mRNA encoding for HBV antigens. Naive C57BL/6 mice (n = 6) were dosed via prime/boost strategy using coformulated (open circles) or co-encapsulated (closed circles) mRNA encoding for the three HBV antigens using a total of 15 µg. (**a**) Induced immune responses, as measured by IFN-γ ELISpot (vertical axis), split by peptide stimulation (horizontal axis). Levels of induced anti-HBs (**b**) and anti-HBe (**c**) antibodies (vertical axis) in the serum measured via CLIA one day −1, 14, and 28 (horizontal axis). % of polyfunctional CD8^+^ T cells (**d**) and CD4^+^ T cells (**e**) defined as % of IFN-γ and TNF-α double positive cells of parent (vertical axis), split per peptide stimulation (horizontal axis). Results are shown as mean ± standard deviation. Statistical comparison is performed by unpaired *t*-test or one-way ANOVA on log_10_ transformed data for ELISpot and antibodies, and on logit transformed data for ICS (* *p* ≤ 0.05).

**Figure 5 vaccines-12-00237-f005:**
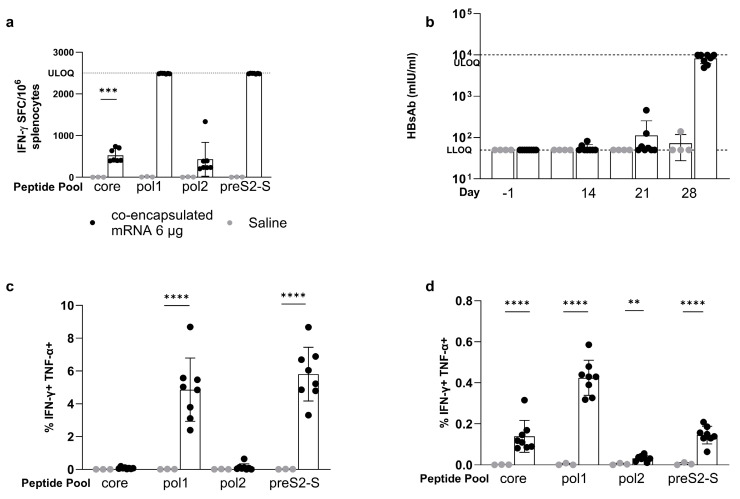
Evaluation of induced immune responses in naive BALB/c mice after vaccination with ALC-0315-co-encapsulated mRNA encoding for HBV antigens. Naive BALB/c mice were dosed via a prime/boost strategy using co-encapsulated mRNA encoding for the three HBV antigens using 6 µg (black circles, n = 8) and were compared to saline control (grey circles; n = 4). (**a**) Induced immune responses, as measured by IFN-γ ELISpot (vertical axis), split by peptide stimulation (horizontal axis). (**b**) Induced concentrations of anti-HBs (vertical axis), measured by CLIA, on day −1, 14, 21, and 28 (horizontal axis). % of polyfunctional CD8^+^ T cells (**c**) and CD4^+^ T cells (**d**) defined as % of IFN-γ and TNF-α double positive cells of parent (vertical axis), split per peptide stimulation (horizontal axis). Results are shown as mean ± standard deviation. Statistical comparison is performed by unpaired t-test or one-way ANOVA on log_10_ transformed data for ELISpot and antibodies, and on logit transformed data for ICS (** *p* ≤ 0.01, *** *p* ≤ 0.001, and **** for *p* ≤ 0.0001).

**Figure 6 vaccines-12-00237-f006:**
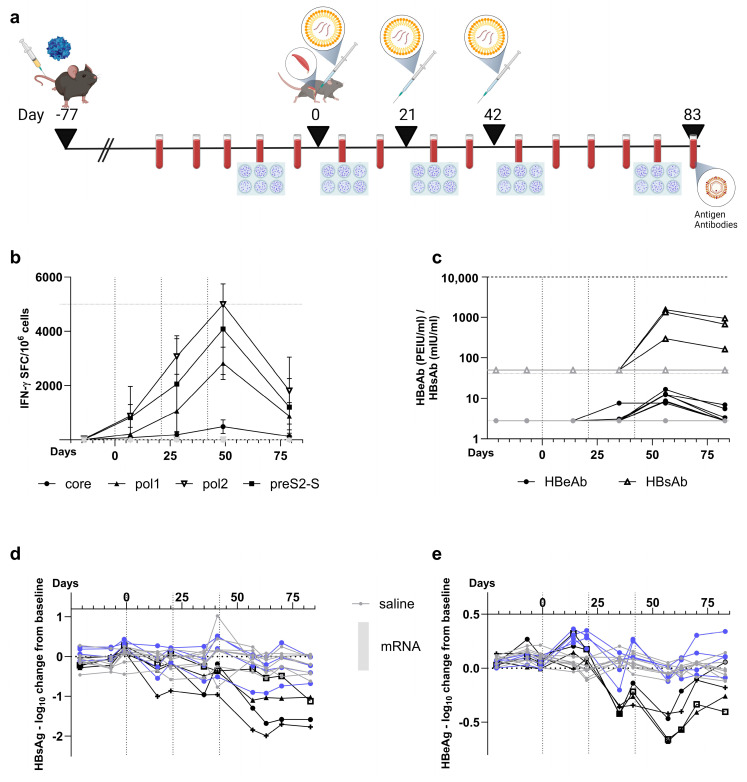
**Evaluation of ALC-0315-co-encapsulated mRNA vaccine in AAV-HBV-transduced C57BL/6 mice**. (**a**) Scheme of AAV-HBV-transduced mouse experiment where the mice were dosed 77 days after transduction (n = 8). Vaccination with co-encapsulated mRNA vaccine occurred three times with 21-day intervals where the first dose was set on day 0. Blood was drawn weekly (red tubes), and on day −14, 7, 28, 49, and 79 post dosing, leucocytes were isolated for IFN-γ ELISpot (plates). On all other weeks, HBsAg, HBeAg, anti-HBs, and anti-HBe were measured. Mice were vaccinated at 10 µg (black) compared to saline control (grey). (created with BioRender.com, accessed on 19 January 2024) (**b**) Induced immune responses (mean ± SD) over time (horizontal axis), as measured by IFN-γ ELISpot (vertical axis), split per peptide stimulation (core: circles; pol1: triangles; pol2: reverse open triangles; and preS2-S, squares). (**c**) Induced concentrations (vertical axis) of anti-HBs (triangles) or anti-HBe (circles) antibodies, measured via CLIA for vaccinated (black) and saline control (grey) shown for each individual mouse. Log_10_ change from baseline (vertical axis) for concentrations of HBsAg (**d**) and HBeAg (**e**) measured via CLIA for mRNA-vaccinated mice (blue circles for the mice without effect on HBsAg levels and black for the mice with >1 log_10_ reduction in HBsAg levels, for which each mouse is visualized with a different symbol that reoccurs in subsequent figures); and saline control (grey) shown for each individual mouse.

**Figure 7 vaccines-12-00237-f007:**
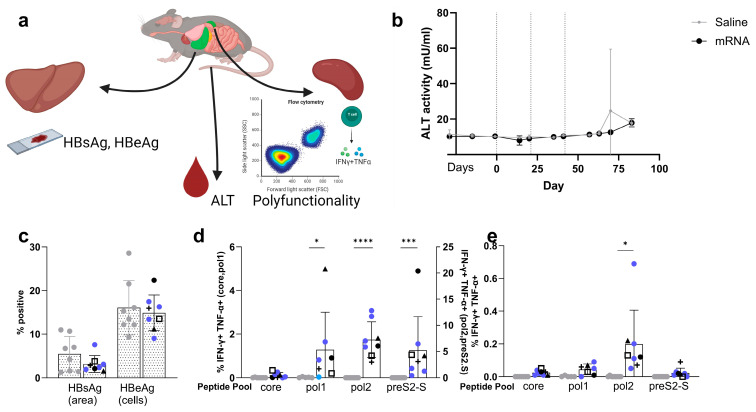
**Evaluation of liver and spleen at the end point of the AAV-HBV-transduced mouse experiment.** (**a**) Schematic overview of performed assays on liver and spleen at the end of the efficacy study in AAV-HBV-transduced mice. Flow cytometry was performed on purified splenocytes, ALT-levels on serum, and viral parameters were assessed on FFPE slides of liver. The spleen was used to check for polyfunctional T cells (created with BioRender.com, accessed on 24 February 2024). (**b**) ALT levels measured over time in the AAV-HBV experiment. (**c**) % of the liver area positive of HBsAg and the proportion of HBcAg-positive cells (vertical axis) for vaccinated (black and blue; corresponding to Figure 6d,e) and saline control group (grey) measured using immunohistochemistry on FFPE slides from the liver (dotted bars). % of polyfunctional CD8^+^ T cells (**d**) and CD4^+^ T cells (**e**) defined as % of IFN-γ and TNF-α double positive cells of parent (vertical axis) in spleen, split by stimulation peptide (horizontal axis). For (**c**–**e**): blue circles are used for the mice without any effect on HBsAg levels and black for the mice with >1 log_10_ reduction in HBsAg levels, for which each mouse is visualized with a different symbol corresponding to the symbols in Figure 6d,e); saline control in grey. Results are shown as mean ± standard deviation. Statistical comparison is performed by unpaired *t*-test or one-way ANOVA on logit transformed data (* *p* ≤ 0.05, *** *p* ≤ 0.001, and **** for *p* ≤ 0.0001).

## Data Availability

The datasets used and/or analyzed during the current study are available from the corresponding author on reasonable request.

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
