# Peer review of "mRNA Therapeutic Vaccine for Hepatitis B Demonstrates Immunogenicity and Efficacy in the AAV-HBV Mouse Model"

_vaccines, 2024, doi:10.3390/vaccines12030237_

Round 1

Reviewer 1 Report

Comments and Suggestions for Authors

The manuscript by de Pooter et. al. investigated the immunogenicity of mRNA vaccines targeting HBV core, polymerase and Pre-S2 and S antigens in naïve mice and the aav-HBV mouse model. The authors tested individual and co-formulations in a prime-boost strategy and measured HBV-specific T cell immunity using elispot and intracellular staining and antibodies to HBs and HBe. The mRNA vaccines were able to induce T cells and anti-HBs after prime/boost. The authors added a second boost in the aav-HBV mouse model and observed increasing magnitude of immunity from the first boost at 21d. They observed >1 Log10 HBsAg decline in 4/8 mice after the second boost. However, HBeAg rebounded in the mice, suggesting minimal hepatocyte clearance had occurred.

Testing of lipid nanoparticle-mRNA vaccines in CHB is logical given their success in SARS-CoV-2 protection and relative ease of production. The first part of the manuscript tests different formulations of the mRNAs but lack a robust comparison between the experiments to demonstrate the benefit/non-inferiority of co-formulation. Experiments performed in the AAV-HBV mouse model lacked mechanistic detail to explain the HBsAg reduction or checkpoint inhibitor expression on HBV-specific T cells that should accompany mouse model experiments. Therefore, the overall manuscript is relatively observational. The results text could also be improved with more specific wording to address the transparency of the significance, or lack thereof, of different T cell responses in different experiments. Specific comments below.

Specific comments

In figure 2C, lines 292 – 294, the wording gives the impression that core-specific responses were induced but the core response was not significantly increased after prime+boost. Increasing the amount of mRNA did not change that. The authors should reconsider the wording.

I would make the same comment regarding the specificity of wording for both Pol and HBs responses. While I agree the responses look increased in the prime+boost, the authors should be more transparent about the significance of the results in the text.

Figure 3 lacks a comparison between individually formulated mRNAs and co-formulated mRNAs to determine whether co-formulation is good, bad, or neutral with regards to T cell/antibody induction.

The authors did not provide any rationale for adding an additional boost in the aav-HBV mouse model. This presumably because the responses were too weak with a single boost. This data could be shown, or at least the addition of another boost explained in the text.

The reduction of HBsAg cells was not significant and HBeAg+ cells remained unchanged.  

Was the HBsAg reduction in aav-HBV mice in Fig 6E related to antibody immune-complex formation that correlates with the magnitude of anti-HBs or antiviral effects of T cells? In addition, representative histology images for HBs and HBc staining should be included.

Was there any evidence of clearance of infected hepatocytes, such as elevated ALT?

Staining for checkpoint inhibitors on the total CD4/CD8 T cell populations is not informative. This should be done with well-established MHC multimer for HBV-specific responses in mice to represent the effect on HBV-specific T cells.

In the last paragraph, line 514 the authors make no mention of the lack of HBV-specific T cells in the liver. Again, relating to my comment on transparency in the text accurately reflecting data in figures.  

The authors discuss the weak core responses in the aav-HBV model (lines 559 – 568) but not the distribution of core response in naïve animals. Core is typically the most immunogenic antigen in chronic hepatitis B patients. In each model tested, core-specific T cells were the weakest responses. Can the authors speculate in the discussion why this might be?

Reviewer 2 Report

Comments and Suggestions for Authors

In this study, De Pooter and colleagues developed lipid nanoparticle formulated trivalent mRNA therapeutic vaccine against chronic hepatitis B virus (HBV) infection. The authors in details examined the immunogenicity of the vaccine in HBV-naïve BL/6 and balb/c mice and studied the antiviral efficacy in relevant mouse model, AAV-HBV. They demonstrate that the vaccine induces HBV-specific antibody response and polyfunctional CD4 and CD8 T cell responses, and reduces HBV antigens in circulation, albeit transiently, in half of the immunized HBV carrier mice. The authors foresee combining their vaccine in a multimodal therapeutic regimen to further improve its performance. The work presents a significant contribution in the field of therapeutic HBV vaccination, which represents novel and promising strategy to treat hepatitis B. Although the manuscript is well-written, the experiments are well-designed and interesting, and the results overall support the author’s conclusion, there are several issues and limitations that could be addressed and would improve the manuscript. 

1.    In the abstract, when describing the results of the study in AAV-HBV mice, the authors describe the result for the best responder (1.7log10 reduction in HBsAg and 50% reduction in HBeAg), while only half of the mice did not demonstrate significant decrease in serum HBV antigen levels, there was no significant reduction in HBV antigens in the liver, and even in the responding mice the effect on HBeAg was overall transient. The statement in the abstract might be therefore misleading and should be carefully revised to reveal the outcomes of the study more objectively.

2.    Supplementary Figures S4 and S5, although important, there is no citation in the text.

3.    The genotype of HBV used in AAV-HBV is not mentioned. Since the sequences of mRNA vaccines were optimized for various HBV genotypes, it is important to mention against which HBV genotype the efficacy was examined.

4.    In flow cytometry studies, low frequencies of polyfunctional IFNg/TNF-positive CD4 and CD8 cells were detected, mostly in AAV-HBV model. Information on how many lymphocytes were acquired on average would be useful.

5.    In the first chapter of the results, when authors describe immunogenicity study using individually formulated mRNA vaccines, authors should clearly specifiy which vaccine they are using, since this information is missing in the text. In the paragraph starting in line 292, I guess it is mRNA encoding HBV core, in line 314 mRNA encoding HBV pol and in line 327 mRNA encoding preS2-S, still, I guess…

6.    In Supplementary Figure S1 the authors present anti-HBe antibodies detected in HBV-naïve BL/6 mice. Since ‘e’ antigen was not included in the vaccine, nor the mice were infected with HBV, can authors explain why these antibodies were measured and why they were detected?

7.    In the legend of Figure 6C anti-HBs antibodies are marked as circles, and anti-HBe as triangles, but in the Figure 6C it is opposite.

8.    Figure 6C-E shows the results of individual AAV-HBV mice marked in blue, pink and black. I understand the authors’ intentions, but these figures are very hard to read. The authors claim that 8 mice received vaccination, but I can only find 6 vaccinated mice at best (in Figure 6C, only 4 for anti-HBsAb). The authors should consider their choice of color and symbols, and revise the Figures, to better distinguish between the individual mice, or at least consider marking the immunized mice which did not reduce serum HBeAg and HBsAg in one style? I believe that this will help the reader to easier correlate the changes in different parameters in the individual mice, mostly in these with the strongest antiviral effect.

9.    In Figure 6D-E only the fold changes in circulating HBV antigens are shown. Why do authors not show the original values? Was there any difference in the initial levels of the HBV antigens between the mice who responded to vaccination vs the ones who did not?

10.  Why in Figure 7 the individual mice are not highlihted with the individual colors/symbols as in Figure 6? It would be interesting to compare whether the responder mice showed higher expression of checkpoint markers or more polyfunctional cells.

11.  In Figure 7C-D the authors show the frequencies of immune checkpoint markers on total CD4 and CD8 T cells from liver of the AAV-HBV mice. The authors claim that the changes (in the majority, not significant) indicate ‘that vaccination induced activation of T cells’. Without analyzing antigen-specific cells these data are hard to interpret. This limitation of the study should be discussed.

Comments on the Quality of English Language

 Authors should carefully revise the manuscript again for minor spelling and stylistic mistakes. I just found a few. In line 485 stands HBcore but the authors mean HBeAb, line 593 - HBsAg is misspelled, in line 41 can’t should be corrected to cannot; line 334 in the case of pol – should be changed for: except for pol, line 339: singly – for: individually, etc…

Round 2

Reviewer 1 Report

Comments and Suggestions for Authors

The authors have addressed my primary comments by improving the transparency of the results text and adding important data that helps with interpretation.

There is significant interest in testing mRNA vaccines in CHB patients and a big gap between these mouse models and humans. I understand the limitations.

Author Response

Dear

Many thanks for your valuable input. We acknowledge that the paper has improved based on your comments. We are glad that we have been able to address all of your comments them to your satisfaction.

Yours Sincerely,

Dorien De Pooter, on behalf of all co-authors